# The Rhizosphere Microbiome of Ginseng

**DOI:** 10.3390/microorganisms10061152

**Published:** 2022-06-02

**Authors:** Paul H. Goodwin

**Affiliations:** School of Environmental Sciences, University of Guelph, 50 Stone Road East, Guelph, ON N1G 2W1, Canada; pgoodwin@uoguelph.ca

**Keywords:** bacteria, cultivation, fungi, *Panax*, soil, stress, symbioses

## Abstract

The rhizosphere of ginseng contains a wide range of microorganisms that can have beneficial or harmful effects on the plant. Root exudates of ginseng, particularly ginsenosides and phenolic acids, appear to select for particular microbial populations through their stimulatory and inhibitory activities, which may account for the similarities between the rhizosphere microbiomes of different cultivated species of *Panax*. Many practices of cultivation attempt to mimic the natural conditions of ginseng as an understory plant in hilly forested areas. However, these practices are often disruptive to soil, and thus the soil microbiome differs between wild and cultivated ginseng. Changes in the microbiome during cultivation can be harmful as they have been associated with negative changes of the soil physiochemistry as well as the promotion of plant diseases. However, isolation of a number of beneficial microbes from the ginseng rhizosphere indicates that many have the potential to improve ginseng production. The application of high-throughput sequencing to study the rhizosphere microbiome of ginseng grown under a variety of conditions continues to greatly expand our knowledge of the diversity and abundance of those organisms as well as their impacts of cultivation. While there is much more to be learnt, many aspects of the ginseng rhizosphere microbiome have already been revealed.

## 1. Introduction

The main environments occupied by microorganisms associated with plants can be divided into the endosphere inside plant tissues, anthosphere inside flowers, spermosphere inside seeds, carposphere inside fruits, phyllosphere on the leaf surface, and rhizosphere in soil adjacent to roots [1]. This review will concentrate on the rhizosphere, which has the highest numbers of microorganisms among those environments and can have significant impacts on plant growth and health [2]. Although not covered in this review, endosphere microbes, particularly in roots, are often rhizosphere microbes that enter through wounds, such as those formed by lateral root formation, by natural openings such as root tips, or by directly penetrating the root epidermis using extracellular cell wall degrading enzymes [3].

The rhizosphere is distinguished from bulk soil by having a much greater level of microbial activity [3]. Traditionally, microbes in the rhizosphere were studied using cultivation-based methods, but the ability to grow on standard culture media favors only a fraction of the microbial community [4]. Thus, different DNA-based culture-independent methods have been developed [5]. An example would be the use of polymerase chain reaction denaturing gradient gel electrophoresis (PCR-DGGE), where DNA is extracted and PCR products for sequences, such as 16S or 18S ribosomal RNA, which have similar lengths but different nucleotide sequences, can be separated on a gel with a gradient of DNA denaturing agents, allowing the partially melted DNA to have different electrophoretic mobilities. Microarrays are another approach, where extracted DNA is labeled with a fluorophore and hybridized to known microbial sequences on a surface that is detected by fluorescence microscopy. With fluorescence in situ hybridization (FISH), microbes are collected on membrane filters, then DNA is released and hybridization to fluorescent-labeled DNA probes is detected by image analysis. However, those techniques have limitations, such as the difficulty of PCR-DGGE to separate PCR products with highly similar sequences, reliance on using known microbial sequences and cross-hybridizations with microarrays, and a similar reliance on known microbial sequences and low sensitivity with FISH [6]. More often now, the DNA-based culture-independent method utilized is high-throughput sequencing, including rhizosphere soil [6]. Sequencing 16S ribosomal RNA is typically used for bacteria and archaea, whereas sequencing 18S ribosomal RNA and/or internal transcribed spacer (ITS) regions are most often used for fungi. The sequences are then grouped, such as into Operational Taxonomic Units (OTUs), using microbial sequence databases, typically based on >97% similarity [7,8]. High-throughput sequencing can detect much higher levels of diversity compared to culture-dependent techniques. For example, there were 60% more bacterial OTUs identified by high-throughput sequencing than bacterial species cultured from the maize rhizosphere [9]. However, isolation still has the advantage of producing pure cultures for laboratory studies, such as determining various biological activities.

Rhizosphere microbial diversity and abundance can be affected by many factors, but most studies examine the effects of the rhizosphere environment or plant genotype. An example of the impact of the soil is a study showing that the rhizosphere bacterial communities of lettuce were similar in diluvial sand and alluvial loam but different from those in loess loam, likely due to a stronger rhizosphere effect in diluvial sand and alluvial loam [10]. Another example is that high organic matter in soils of watermelon resulted in higher abundance of beneficial bacteria and higher bacterial diversity than without added organic matter [11]. An example of a study on the plant genotype effect is a higher rhizosphere microbial diversity and abundance found in *Cardamine hirsuta* compared to related *Arabidopsis* species believed to be due to differences in root exudates [12]. 

A major reason for studying rhizosphere microorganisms is that many can benefit plants, such as by enhancing soil aggregate formation, thus improving water retention, aeration, and movement of nutrients and water [13]. Other benefits from rhizosphere microorganisms are increasing plant nutrient availability, such as by solubilizing phosphate (P), fixing nitrogen (N_2_), decaying organic matter, and reducing plant diseases through direct antagonism against pathogens and/or induction of host PAMP-triggering immunity (PTI) [14,15]. Rhizosphere microorganisms can even directly promote plant growth by secreting plant hormone-like compounds or lowering inhibitory levels of plant hormones [14,16].

Compared to model plants or major annual field crops, the rhizosphere microbiology of many horticultural crops, like *Panax* species, are less well studied. However, as this review shows, the number of studies on the ginseng rhizosphere microbiome has greatly increased in recent years, particularly for Asian ginseng (*Panax ginseng*), American ginseng (*Panax quinquefolius*), and Himalayan ginseng (*Panax notoginseng*).

## 2. Ginseng Rhizosphere Environment

*Panax ginseng* is native to the hills and mountains of Korea, northern China, and western Siberia, found in mixed hardwood forests of well-drained sandy loam soils with a pH of 5–6.5 [17]. *Panax quinquefolius* prefers very similar conditions in the hills and mountains of eastern North America from Maine to Minnesota and Ontario to Georgia [17]. *Panax notoginseng* also prefers similar conditions but has a limited range, being native to the forests of the Wenshan mountains of Yunnan province [18]. Ginseng plants can grow 30 or more years in the wild [19]. This indicates that wild ginseng would have a relatively stable rhizosphere in soil with a neutral to acidic pH, good aeration, and annual inputs of nutrients from ginseng root exudates and relatively large amounts of leaf litter from the forest canopy. Typically, perennial plants accumulate root exudates in the rhizosphere at higher concentrations than annual plants due to several seasons of growth without disruption of the rhizosphere by harvesting, allowing for greater root tissue density and mass [20]. Compared to acidic or basic soils, near-neutral pH soils generally have a greater diversity and abundance of microorganisms in the rhizosphere [21]. Good soil aeration promotes water, oxygen, and nutrient diffusion, enhancing microbial activity [22]. Forest leaf litter adds organic carbon to the upper soil horizon, unlike annual crops, where limited amounts of organic matter are typically added or remain after harvest [23]. The significance of forest leaf litter to ginseng was observed when leaves from five tree species were added to soil of *P. ginseng* resulting in higher total nitrogen, available nitrogen–phosphate–potassium (NPK), carbon/nitrogen ratio (C/N), and soil microbial biomass, particularly, increased numbers of Proteobacteria [24]. Thus, the rhizosphere of wild ginseng in a forest understory should be relatively rich in organic matter with good oxygen and water exchange that should promote the growth of rhizosphere microorganisms.

Ginseng is also cultivated under relatively similar conditions in cool temperate regions of the world. Common production methods begin prior to transplanting with soil fumigation using broad spectrum pesticides to reduce populations of soil-borne pests, including pathogenic fungi [19]. This would also dramatically reduce the microbiome in the soil [25]. If required, soil pH is adjusted to 5.5–6.5, and fertilizers are applied where the major benefits come from NPK, as well as calcium and magnesium, that have been shown to significantly increase root yield and ginsenoside content [26]. After fumigation, the soil is tilled to form raised beds to increase water drainage [19]. Tillage typically reduces soil organic matter [27]. Seeds are planted in the fall or one-year old seedlings are transplanted in the spring and then covered with straw mulch acting like leaf litter in the wild, thus adding organic matter. However, there would not be inputs from forest leaf litter as shading occurs artificially by the use of polypropylene mesh shades [19]. Periodic pesticide applications to control insects, pathogens and weeds would have a negative impact on soil microbes. Harvesting of plants after 3–4 years involves tillage and root removal that would result in reduced organic matter negatively impacting the soil microbiome. Thus, compared to wild ginseng, cultivated ginseng would start with a highly disrupted soil with relatively little soil organic matter and reduced soil microbial populations. A rhizosphere would develop over the several years of growth from ginseng root exudates, mucilage, and root cap cells that might regularly be reduced by pesticide inputs but increased by organic inputs from decaying straw mulch and senescent ginseng stems and leaves that dieback each fall. Thus, the rhizosphere of cultivated ginseng would be expected to have significant differences from that of ginseng in the understory of a mixed hardwood forest, starting with less microbial diversity and abundant microbial population but possibly more affected by cultivation practices over the years of growth. 

Part of the rhizosphere effect of ginseng would be ginsenosides released into the soil in the root exudates, estimated at 25 µg per day per plant of *P. quinquefolius* [28]. The specific ginsenoside compounds in the exudates were F11, Rb1, Rb2, Rc, Rd, Re, and Rg1, which were the same as those found in roots. For *P. notoginseng*, the specific ginsenoside compounds were Rg1, Rb1, and Rd in root exudates, which were released at 1.2, 1.0, and 2.6 µg per day per plant, respectively [29]. Another type of ginseng root exudate is phenolic compounds, such as benzoic acid, diisobutyl phthalate, diisobutyl succinate, palmitic acid, and 2,2-bis-(4-hydroxyphenyl) propane, that have been detected in rhizosphere soil of *P. ginseng* [30]. The phenolics, gallic acid, salicylic acid, 3-phenylpropionic acid, and cinnamic acid, were also detected in the rhizosphere of cultivated *P. ginseng* [31]. The exudates of *P. ginseng* roots are complex with over 30 compounds composed of 23% hydrocarbons, 64% organic acids/esters, 13% phenolic acids, and trace amounts of ginsenosides [32]. However, ginseng root exudates can be altered by different factors, such as N and K deficiency, which resulted in more types of organic acids and phenolics released compared to P deficiency, which resulted in less [33]. 

## 3. Ginseng Rhizobiome Diversity and Abundance

Several reports have described the diversity and abundance of bacteria and fungi in the rhizosphere of *P. notoginseng* [34,35,36,37,38]. All those studies examined *P. notoginseng* under cultivation using high-throughput sequencing, but Wu et al. [34] examined 2-to-4-year soil collected in winter, Fan et al. [35] examined 3-year-old soil collected in summer, Tan et al. [36] and Tan et al. [37] examined 1-to-3-year old soil collected in fall, and Miao et al. [38] examined 3-year-old soil collected in spring. Thus, samples were collected from different aged roots at different times of the year, and the soil microbiome is known to be affected by differences in temperature and moisture [39]. Despite such differences, there were a number of similarities between the studies. Proteobacteria and Actinobacteria were in the three most abundant bacterial phyla, whereas Bacteroidetes, Firmicutes, and Acidobacteria were also mentioned among the most common phyla (Table 1). Among the Proteobacteria, *Pseudomonas* was listed as a common genus, and *Arthrobacter* and *Streptomyces* more often noted as abundant among the Actinobacteria. For fungi, the Ascomycota, Basidiomycota, and Zygomycota were the most abundant phyla, but ranked differently between the studies, although Basidiomycota was never the most abundant (Table 1). Commonly mentioned abundant genera were *Fusarium* in the Ascomycota, *Cryptococcus* in the Basidiomycota, and *Mortierella* in the Zygomycota. 

There are also multiple reports about the diversity and abundance of bacteria and fungi in the rhizosphere of *P. ginseng* [40,41,42,43,44]. The methods in the studies were PCR-DGGE [40], RAPDs [42], and high-throughput sequencing [41,43,44]. All the studies collected samples in the spring, except Fang et al. [43], who collected in late summer. Plant ages in the soil differed with 1- to 4-year-old gardens [40], 1- to 6-year-old gardens [42], 2- to 6-year-old gardens [41], 1- to 5-year-old gardens [44], and 15-year-old gardens [43]. However, there were still many similarities in the results. For rhizospheric bacteria, the dominant phylum was typically the Proteobacteria, with *Pseudomonas* mentioned among the most abundant genera in that phylum (Table 2). The other abundant phyla varied between the studies, with Actinobacteria, Firmicutes, Acidobacteria, and Chloroflexi in more than one study, whereas Fusobacteria was only noted in one study [40]. For rhizospheric fungi of *P. ginseng*, the most abundant phylum was the Ascomycota with *Fusarium* mentioned as the most abundant in that phylum in two studies (Table 2). This was followed by Basidiomycota or Zygomycota with *Mortierella* noted as abundant in the Zygomycota. 

The other *Panax* species with multiple reports about the diversity and abundance of bacteria and fungi in the rhizosphere is *P. quinquefolius* [45,46,47,48]. In all these studies, the rhizobiome was examined with high-throughput sequencing using 3- and 4-year-old garden rhizosphere soils, but they differed in collection times, with Jiang et al. [45] sampling in spring, Dong et al. [47] sampling in summer, and both Samur [48] and Liu et al. [46] sampling in fall. For rhizosphere bacteria, most studies showed that Proteobacteria was the most abundant phylum followed by Actinobacteria or Acidobacteria, although the ranking differed between studies (Table 3). Only Dong et al. [47] did not find that Proteobacteria was the most abundant phylum. *Rhodoplanes* was most often noted as an abundant genus for the Proteobacteria, *Arthrobacter* for the Actinobacteria, and *Candidatus solibacter* for the Acidobacteria. For the rhizosphere fungi of *P. quinquefolius*, the dominant phylum was the Ascomycota in all studies, with *Penicillium* often noted as an abundant ascomycete genus (Table 3). The Zygomycota with *Mortierella* was noted in two studies, while the Basidiomycota, Rozellomycota, and Chytridiomycota were also abundant phyla in one study each. 

Comparing between *Panax* species, it appears that they have similarities in their rhizosphere microbiomes (Table 1, Table 2 and Table 3). At the phylum level, the rhizosphere of *P. notoginseng*, *P. ginseng*, and *P. quinquefolius* most often had Proteobacteria being dominant followed by either Acidobacteria or Actinobacteria. There was more similarity between *P. notoginseng* and *P. ginseng*, with *Pseudomonas* and *Burkholderia* commonly being dominant compared to *P. quinquefolius*, with *Rhodoplanes* and *Candidatus solibacter* being most dominant. A *Panax* genotype affect could possibly be seen in *Rhodoplanes* being the most common for *P. quinquefolius* in both China and North America, while it was not common for either of the other two *Panax* species. Studies of rhizosphere fungi of *P. notoginseng*, *P. ginseng*, and *P. quinquefolius* almost always showed that the Ascomycota was the most common phylum, followed by Basidiomycota and then Zygomycota. For all three *Panax* species, *Fusariums* and *Mortierella* tend to be among the most common fungal genera in the rhizosphere. 

## 4. Impact of Cultivation of the Rhizosphere Microbiome

Many ginseng cultivation practices can affect the rhizosphere microbiome. Chloropicrin is a potent broad-spectrum pesticide against microbes, weeds, insects, and nematodes, and fumigation of ginseng soil with it reduced levels of *Pythium* by 66% one year after fumigation [49]. Thus, it is likely that many other microbes were also killed by the fumigation. Repeated exposure to fungicides during cultivation was proposed to explain why fungal endophyte diversity was lower in 4-year-old roots than in 1-, 2-, and 3-year-old ginseng roots [50]. Thus, pesticides also likely reduced microbial diversity in the rhizosphere, considering that many root endophytes originate from the rhizosphere [51]. However, not using fungicides could result in higher amounts of disease that would also affect the microbiome. *Panax ginseng* plants with more disease were associated with increases in Acidobacteria, Alphaproteobacteria, and Solibacteres in the rhizosphere [41]. While there are no reports of chemical fertilization affecting the microbiome of ginseng, studies in other crops like maize have shown that higher N due to urea application increased root exudates and thus, soil bacteria abundance [52]. Mulching clearly affects the rhizosphere microbiome of ginseng. The numbers of bacteria in the Bacteroidetes were more reduced in *P. ginseng* soil than were other bacteria when conifer leaf litter mulch was applied versus broad leaf litter mulch [24]. White and red pine bark mulch reduced root infections of *P. quinquefolius* by *Rhizoctonia solani* compared to oat straw mulch, possibly due to the bark leachates releasing an enteric bacterium and a yeast that were inhibitory to *R. solani* [49]. Soil additives, like vermicompost, a mixture of compost and worm castings, reduced the level of Fusarium root rot and increased the relative abundance of the beneficial bacteria *Pseudomonas*, *Lysobacter*, and *Chryseolinea* in the rhizosphere of *P. quinquefolius* [53]. Artificial shading used in ginseng gardens would likely have an effect as it not only reduces light intensity but also reduces air temperatures over ginseng beds up to 6 °C [54]. Even the type of shading appears to have an impact. Rhizosphere bacterial diversity of *P. quinquefolius*, notably for the Proteobacteria and Nitrospirae, was more abundant using 2.6 m high shades with an arched shape compared to 1.6 and 2.0 m high shades with a flat shape [55]. This may be related to different soil physicochemical properties associated with the different shade types, where the soil pH was significantly higher in the arched shades and increased soil pH was positively correlated with microbial diversity. Soil total organic matter, hydrolyzable nitrogen, and available phosphorus were also affected by shading. Disease incidence and severity were lower and the yield was higher with flat shades compared to the arched shades. Possibly the arched shades allowed for higher light levels, but as this was not measured, it is not clear why the type of shadings had different impacts. Evidence for an effect of harvesting comes from an examination of the bacterial and fungal microbiome of ginseng soil following harvesting where the soil was dug to 30 cm and the roots harvested by hand [48]. There were major shifts in diversity and abundance of the microbiome, such as increased abundance of Proteobacteria and Acidobacteria and decreased abundance of Actinobacteria by the following spring after harvest, compared to other phyla, which initially fluctuated in abundance but returned to comparable levels in the following year. Changes in the microbiomes due to harvesting were proposed to result from soil disruption by tillage and the addition of organic matter to the soil from ginseng crop debris decaying after harvest. Thus, it appears that ginseng cultivation practices can directly or indirectly impact the microbiome of the rhizosphere.

Changes in the soil microbiome as a crop develops are determined by both changes in the physiology of the plants as well as different management practices applied during the growing season. As ginseng is a perennial, the microbiome could be affected over the years of growth as root biomass increases with more exudates being released. This would likely be most obvious in cultivated ginseng gardens over the first several years, when growth is fastest and the rhizosphere is being established.

During 3 years of farmland cultivation of the first crop of *P. ginseng*, soil bacterial diversity and abundance decreased with the greatest decreases in the Proteobacteria and Bacteriodetes, possibly due to older roots excreting less organic material [41]. The second crop (i.e., replanted ginseng) had a higher abundance of Acidobacteria, Solibacteres, and *Rhizomicrobium* in the soil compared to the first crop soil and non-cultivated soil, which was correlated with decreased soil pH caused by long-term ginseng growth. For farmland cultivation of *P. ginseng*, cultivation age had a more pronounced effect on soil bacteria than on soil fungi with decreasing pH in older soils related to negative impacts on soil bacteria [56]. While overall abundance was higher in 6- than in 5-year-old cultivated soil, beneficial and neutral microbes decreased over time, while potentially harmful microbes, such as the pathogenic fungi, *Cylindrocarpon* and *Fusarium*, increased over time. The authors proposed that higher abundances of pathogens over time led to an imbalance in the microbial community. For farmland cultivation of *P. ginseng*, the greatest changes in rhizophere bacteria were in the first three years, gradually stabilizing in the fourth and fifth years [44]. Fungi were less affected over time than bacteria. However, non-pathogenic fungi decreased over time, whereas pathogenic fungi, such as *Cylindrocarpon* and *Nectria*, increased over time. Tong et al. [44] proposed that allelochemicals from ginseng roots may be increasingly disturbing the microbial community over time. Li et al. [57] found that *P. ginseng* growth over 6 years increased soil fungal abundance primarily for an *unclassified_Nectriaceae*, *Fusarium*, *Gibberella*, *Cryptococcus*, and *Trichoderma*, but decreased fungal diversity, mostly *Mortierella* and *Humicola.* It was believed that certain phenolic acids found in *P. ginseng* root exudates were responsible. 

Farmland soil with continuous cropping would be altered by many factors, such as tillage, pesticides, and chemical fertilizers, compared to forest soils subjected to little or no crop cultural practices. Comparing wild *P. ginseng* in a forest understory to cultivated *P. ginseng* showed that they had different fungal soil communities, which became more different over time in both soil types with increasing abundance of fungal pathogens but decreasing fungal diversity [58]. Tong et al. [44] found that there was greater richness and diversity of rhizospheric bacteria but not fungi for planted *P. ginseng* in forest versus farmland soil. In addition, after five years of growth, the richness and diversity of soil bacteria increased in both forest and farmland gardens, but the richness and diversity of soil fungi decreased in farmland soil and was unchanged in forest soil. The amount of soil fungal pathogens changed over time in both soils but differed depending upon the fungus. For example, over the years with *P. ginseng* cultivation, *I. mors-panacis* increased in both farmland and forest soil, *Ilyonectria robusta* increased more in forest than farmland soil, and *Fusarium solani* increased more in farmland than forest soil.

Changes in bacterial abundance were observed in the soil of *P. quinquefolius* over 3 years’ growth in the first crop, with significant decreases in levels of *Paralcligenes* and *Sphingomonas* belonging to Proteobacteria, as well as Candidatus Saccharibacteria, but levels of other Proteobacteria, such as *Pandoraea*, as well as Chlamydiae, increased with plant age [47]. There was also increased fungal abundance over time with the greatest increases in the abundance of *Fusarium* and members of the Tremellomycetes, Chytridiomycota, and Sordariales, but decreasing abundance of the Zygomycota. Unlike many other studies of ginseng cultivation, significant soil pH changes over the years were not observed. Additionally, for *P. quinquefolius*, the abundance of soil bacteria with biodegradative functions, such as species of *Methylibium*, *Sphingomonas*, *Variovorax*, and *Rubrivivax*, was lowered over 4 years of growth compared to non-cropped soils, which was accompanied by a decrease in soil pH over time [59]. There was also an increased abundance of the soil-borne fungal pathogens, *Fusarium* and *Ilyonectria*, over years of cultivation relative to non-cropped soils. It was proposed that this change could result in toxic soil compounds accumulating due to reduced microbial degradation. 

In general, these studies reveal that cultivation of *P. ginseng* and *P. quinquefolius* affects the rhizosphere microbiome over time. Although forest and farmland soils may begin differently, these changes occur both in forest and farmland soils as the crop develops. The effects appear to be related to changes in soil factors, such as decreased pH and increased toxic compounds. As a result, the impact is negative for the crop, typically with decreases in potentially beneficial microbes and increases in potentially pathogenic microbes over the years of cultivation. However, it is not clear whether increased abundance of pathogens is a cause or a result of changes in other microbes in the soil. 

## 5. Impacts of Ginseng on Rhizosphere Microbiome

The genotype of ginseng appears to affect the rhizosphere microbiome. Wang et al. [60] reported that the fungal micobiome was more affected than the bacterial microbiome when comparing the *P. ginseng* genotypes, Gaoli, common ginseng, Shizhu, and Biantiao. Rhizosphere fungal diversity was negatively correlated with rhizosphere bacterial diversity with Gaoli, common ginseng, and Shizhu. Biantiao had the lowest rhizosphere microbial diversity, which could be related to seeds of common, Gaoli, and Shizhu being planted directly and grown for six years, while Biantiao are grown for three years before being transplanted as seedlings. For bacteria, Proteobacteria, Acidobacteria, and Verrucomicrobia were abundant in the rhizosphere with all ginseng types, but Acidobacteria were more abundant with Gaoli. At the genus level, the abundance of *Bacillus* was higher with Shizhu and Biantiao than common ginseng and Gaoli rhizospheres. For rhizospheric fungi, Basidiomycota, Ascomycota, and Zygomycota were most abundant with all ginseng genotypes, but the Ascomycota were more abundant and Zygomycota less abundant with Shizhu. At the genus level, *Mortierella* was more abundant with Biantiao, and the potential pathogens, *Alternaria alternata* and *Cladosporium*, were less abundant in the rhizosphere of Biantiao. The authors speculated that differences in the amount and composition of root exudates in the different ginseng genotypes may have been responsible. 

The type of ginseng root exudates, such as hydrocarbons, organic acids/esters, phenolic acids, and ginsenosides [32], can have major impacts on the microbiome of the rhizosphere. Various hydrocarbons and organic acids/esters act both as nutrients maintaining the rhizosphere effect and as signals attracting or repelling soil bacteria and fungi [61]. An example would be *P. ginseng* root exudates that were chemotactic to the soft-rot bacterium *Pseudomonas qessardii* in the culture medium [62]. Among the organic acids in ginseng root exudates is citric acid [32], which is known in cucumber root exudates to induce biofilm formation and/or chemotaxis for the beneficial rhizobacteria, *Bacillus amyloliquefaciens*, and *Bacillus subtilis* [63]. A root ginsenoside, Rg1, of *P. notoginseng* that is found in the rhizosphere causes ginseng root cell death, releasing cellobiose and d-galacturonic acid into the soil, and the addition of those compounds with Rg1 to soil increased populations of pathogens like *Ilyonectria* and decreased beneficial microorganisms like *Pseudomonas* and *Streptomyces* [64]. Li et al. [42] concluded that the microbial rhizosphere community of ginseng was different from that of other plants due to the types of compounds in ginseng roots exudates. 

Ginsenoside saponins are important element of ginseng root exudates, and many saponins have selective antimicrobial activity in the soil altering the rhizosphere microbiome [61,65]. Adding the ginsenosides Rg1, Rb1, and Rh1 or a ginsenoside mixture into ginseng soil decreased populations of beneficial fungi, such as *Acremonium*, *Mucor*, and *Ochroconis*, but increased populations of pathogenic fungi, such as *Alternaria*, *Cylindrocarpon*, *Gibberella*, *Phoma*, and *Fusarium* [66]. Growth of *Pythium irregulare*, a root rot pathogen of ginseng, was stimulated by total ginsenosides or Rb1 but inhibited by F2 added to culture media, and dipping ginseng roots into a ginsenoside solution prior to planting delayed infection by *P. irregulare* [67]. *Pythium irregulare* produces extracellular glycosidases able to partially deglycosylate ginsenosides, and the level of deglycosylation activity strongly correlated with the virulence of twelve *P. irregulare* isolates [68]. Ginsenosides in culture media also suppressed growth of the non-pathogenic rhizosphere fungus, *Trichoderma* [28]. Growth of a *Trichoderma* isolate exposed to protopanaxatriol type ginsenosides in culture was inhibited, perhaps related to the very limited ability of the fungus to deglycosylate ginsenosides compared to the root rot pathogens, *I. robusta* and *I. leucospermi* [69]. The growth of many plant pathogenic fungi that do not have ginseng in their host range, *Aspergillus nidulans*, *Cladosporium fulvum*, *Fusarium oxysporum*, *Alternaria solani*, and *A. porri*, were inhibited by both protopanaxadiol and protopanaxatriol type ginsenosides, but only the protopanaxadiol ginsenoside fraction inhibited growth of the ginseng pathogen, *I. mors-panacis*, which may be due to its limited ability to deglycosylate ginsenosides [70]. 

The adaptation of microbes to the ginseng rhizosphere is indicated by the many reports of ginseng soil bacteria and fungi that can deglycosylate ginsenosides, e.g., [71,72,73,74]. Park et al. [75] noted that there is a large variety of ginseng soil microbes with glycosidases that can remove sugars from ginsenosides, including the fungi *Absidia coerulea*, *Acremonium strictum*, *Curvularia lunata*, *Fusarium sacchari*, *Rhizopus stolonifer*, and *Paecilomyces bainier*, and the bacteria *Bacillus megaterium*, *Burkholderia pyrrocinia*, *Caulobacter leidyia*, *Intrasporangium* sp., *Microbacterium* sp., and *Sphingomonas echinoides*. In addition to deglycoslyation reducing the toxicity of ginsenosides, removal of the attached sugars may also benefit microbes by providing a food source. For example, *Pseudomonas plecoglossicida* isolated from *P. quinquefolius* soil appeared to use ginsenosides as a nutrient in culture, resulting in higher populations in minimal medium when root ginsenoside extract was included in the medium [76]. Yousef and Bernards [77] also reported that *P. irregulare* was able to use the sugars hydrolyzed from ginsenosides as carbon source. It was proposed that ginsenosides were the main substances in root exudates driving changes in the soil microbiome of *P. notoginseng* due to the ginsenosides in root exudates acting as a carbon source for some members of the soil microbiome stimulating their growth while inhibiting others [29]. These studies strongly suggest that the microbiome around ginseng roots have enhanced or reduced growth due to ginsenosides released by the roots that is related to their ability to deglycosylate them to detoxify them and/or assimilate sugars attached to them. 

Another major class of ginseng root exudates that appear to impact the soil microbiome are phenolic root exudates. They are believed to have greater impacts on rhizosphere microbes than other exudate compounds, such as sugars and amino acids [78]. Applying the ginseng root phenols, gallic acid, salicylic acid, 3-phenylpropionic acid, and benzoic acid to soil increased fungal richness but decreased fungal diversity in soil continuously cropped with ginseng for 6 years [57]. The phenolic acid soil treatment also enhanced the abundance of plant pathogens, such as *Fusarium*, *Gibberella*, and *Ilyonectria*, which may explain why those compounds increased root rot disease. Another example is the phenols, benzoic acid, diisobutyl phthalate, diisobutyl succinate, palmitic acid, or 2,2-bis-(4-hydroxyphenyl) identified in the *P. ginseng* rhizosphere and root exudates, that were used to treat soil, which also resulted in reduced diversity of bacteria and fungi compared to a control [42]. Adding ferulic acid, syringic acid, p-hydroxybenzoic acid, p-coumaric acid, or vanillic acid to culture medium decreased mycelium growth and spore production of the pathogen *Fusarium oxysporum* [79]. Adding a stem and leaf ginseng extract high in phenolics to the culture medium inhibited the growth of the bacteria, *Bacillus cereus*, *Salmonella enteritidis*, *Escherichia coli*, and *Listeria monocytogenes* [80]. Accumulation of diisobutyl phthalate in *P. ginseng* soil that likely came from root exudates and decaying roots correlated with declines in populations of *Arthrobacter*, *Burkholderia*, *Rhodanobacter*, and *Sphingobacterium* in soil [81]. Thus, many studies have shown that a variety of ginseng phenolic acid root exudates can have major impacts on both rhizosphere communities of fungi and bacteria, mostly by reducing their diversity and in some cases shifting towards greater pathogen abundance.

## 6. Impacts of Rhizosphere Microbiome on Ginseng

One way that microbes in the rhizosphere can benefit ginseng is to enhance the plant’s access to nutrients. Microbes can acquire iron through the secretion of siderophores, which are secondary metabolites that bind iron in the environment and deliver it to cells via specific receptors [82]. Hussein and Joo [83] identified 23 siderophore-producing fungal isolates in the ginseng rhizosphere with a *Penicillium* isolate showing the strongest activity, but isolates of *Aspergillus versicolor*, *Mortierella turficola*, *F. oxysporum*, *Metarhizium anisopliae*, *Rhodosporidium toruloides*, and *Trichoderma harzianum* also produced siderophores. Farh et al. [69] identified 7 bacterial isolates as strong producers and 1 isolate as a weak producer of siderophores out of 12 isolates from *P. ginseng* rhizosphere. Many other ginseng rhizosphere bacteria have been found to produce siderophores, such as the *P. ginseng* rhizosphere bacterium, *Mesorhizobium panacihumi*, whose siderophores bound excess iron to reduce its toxicity, thus providing ginseng with heavy metal resistance [84]. Siderophore production appears to be relatively common among ginseng rhizosphere microorganisms and may help both in scavenging iron when levels are too low or sequestering it when levels are too high. 

Another nutrient whose solubility can be affected by microorganisms is phosphate [85]. Twenty-two strains of bacteria from the rhizosphere of *P. ginseng* had phosphate solubilizing activity in culture, with an isolate of *Burkholderia vietnamiensis* showing the highest activity whose phosphate solubilization ability was related to gluconic acid production [86]. Another *P. ginseng* rhizosphere bacterium, *P. fluorescens* RAF15, showed phosphate solubilizing activity that was also related to gluconic acid production [87]. Farh et al. [69] identified 5 bacterial isolates as strong and two isolates as weak phosphate solubilizers among 12 isolates from the *P. ginseng* rhizosphere. Hussein and Joo [83] obtained 60 bacterial and 14 fungal isolates from the rhizosphere of *P. ginseng* with phosphate solubilization activity. Bacteria were more effective than fungi in phosphate solubilization with the most effective bacteria being *P. fluorescens* and *Azotobacter chroococcum* and the most effective fungus being a *Penicillium* species. Thus, phosphate solubilisation appears to be a widespread trait among rhizosphere bacteria and fungi of ginseng, but the effectiveness of different microbes can vary considerably.

Another commonly reported benefit to ginseng provided by rhizosphere microorganisms is suppression of plant diseases. Rhizosphere bacteria and fungi can reduce plant diseases through direct antagonism of soil-borne pathogens, as well as by triggering resistance in the plant against pathogens, such as by induced systemic resistance (ISR) [88]. In ISR, priming of defenses typically occurs by a treatment prior to infection, resulting in faster and stronger triggered resistance when a pathogen infects [89]. One example of direct antagonism is a *B. subtilis* isolate from *P. ginseng* soil that significantly reduced lesion numbers caused by *Colletotrichum panacicola*, possibly due to direct inhibition of the fungus on ginseng leaves, as it inhibited the fungus in culture likely due to antibiotic production [90]. Out of 113 fungal isolates from rhizosphere soil of *P. notoginseng*, 13 showed possible antibiotic production with zones of inhibition against one or more of the ginseng pathogens, *Alternaria panax*, *F. oxysporum*, *F. solani*, and *Phoma herbarum* [39]. Guo et al. [91] examined 574 bacterial isolates from *P. notoginseng* rhizosphere and found 33 isolates showing inhibition of the fungal pathogens, *Cylindrocarpon didynum*, *F. solani*, *Phytophthora cactorum*, *P. herbarum*, and *R. solani*, in culture most likely related to antibiotic production, such as lipopeptides produced by *B. subtilis*, which was the most antagonistic. Another example is Sun et al. [92] who screened 169 bacterial strains from rhizospheric soil of *P. ginseng* and found that *B. amyloliquefaciens* showed antagonistic activity against *Botrytis cinerea* in culture due to antibiotic production, as well as the ability to reduce the severity of gray mold foliar lesions. However, induction of host resistance is also possible with ginseng rhizosphere microbes. A *B. amyloliquefaciens* strain from soil of wild *P. ginseng* induced resistance against *P. cactorum*, which causes foliar blight and root rot, when applied as soil drench that was related to a significant up-regulation of the ISR-related defense genes, *PgPR10*, *PgPR5*, and *PgCAT* [93]. Although the majority of the work in this area involves examining rhizosphere microbes that directly inhibit pathogens of ginseng in culture and then sometimes testing them on infected plants, there appears to be considerable scope for more studies on inducing resistance mechanisms in ginseng by non-pathogenic soil bacteria and fungi, considering how often this has been shown in other plant species [94]. 

Ginseng plants can benefit from rhizosphere microbes producing phytohormones, such as indole acetic acid (IAA), resulting in increased plant growth [95]. From *P. ginseng* soil, 50 of 105 rhizobacterial isolates and 11 of 17 rhizosphere fungal isolates exhibited IAA production *in vitro* with isolates of *P. fluorescens*, *Rhodobacter*, *Burkholderia*, *Pseudomonas putida*, and *A. chroococcum* producing the highest levels [83]. Twelve bacterial strains were isolated from the rhizosphere of diseased *P. ginseng* roots that were positive for IAA production in the presence of the IAA precursor, L-tryptophan, with the highest production by *Arthrobacter nicotinovorans* [69]. This trait appears to be relatively common among ginseng soil bacteria, with reports including IAA production by *Rhizobium panacihumi*, *Paenibacillus panacihumi*, *M. panacihumi*, *Duganella ginsengisoli*, and others [84,96,97,98]. Although common, IAA production appears to be less prevalent in the rhizosphere microbiome of ginseng than phosphate solubilization, at 65% versus 82% for fungal isolates and 48% versus 57% for bacterial isolates obtained from the ginseng rhizosphere [83]. 

While most research on the benefits of rhizospheric microbes has concentrated on iron and phosphate nutrition, disease control, and auxins, ginseng rhizosphere microbes can provide other benefits to the plant. An example would be *B. amyloliquefaciens* strain TB6 from the rhizosphere of *P. ginseng*, which promoted root growth and possessed 1-aminocyclopropane-1-carboxylic acid (ACC) deaminase activity [99]. ACC deaminase can promote plant growth by cleaving the ethylene precursor ACC, thus lowering levels of stress-related ethylene that can inhibit growth [16]. There are also examples of ginseng rhizosphere microbes that promote ginseng growth where the mechanism has yet to be examined, such as *Brevundimonas terrae*, which promoted *P. ginseng* growth [100].

## 7. Conclusions

Like other plant microbiomes, the microbiome of the ginseng rhizosphere is greatly influenced by both plant genotype and environmental factors. Cultivation practices are a significant factor, as ginseng production involves fumigation, tillage, mulching, pesticides, shading, fertilizing, and harvesting. Many of those practices attempt to mimic the natural environment of ginseng as a cool weather hardwood forest understory plant. Compared to wild ginseng plants, cultivation involves a higher plant density, more chemical inputs, and greater soil disturbances. However, while forest and farmland soils may initially differ in their microbiomes, there are many similarities in the effect of ginseng growth on the rhizosphere microbiome over time. A decrease in soil pH negatively affecting rhizospheric bacteria and a shift in rhizospheric fungi with pathogens becoming more dominant over time appears to be common. Despite studies using different locations, collection times, and methods of assessment, there are many similarities between the types of rhizosphere microbes of *P. notoginseng*, *P. ginseng*, and *P. quinquefolius*, indicating that the complex assortment of compounds exuded by roots of *Panax* species have a unique impact on the microbes around them, selecting for a set of microbes more adapted to those compounds. While high-throughput sequencing has given a better idea about the diversity of the ginseng rhizosphere microbiome, more work is needed on studying microbes isolated from the ginseng rhizosphere. A quick survey of the literature by the author identified over 20 publications describing new bacterial species that were isolated from ginseng soil, mostly in the last 15 years, with little to no description of their effects on the plants. There are many reports of ginseng rhizosphere microbes providing benefits to the plant by a variety of mechanisms that can be exploited to reduce fertilizer and pesticide inputs, so continuing to isolate and screen ginseng soil microbes for their impacts on ginseng is important. It is likely that many more novel ginseng rhizosphere microorganisms, including many with beneficial traits and many new species, will be found. The study of the ginseng rhizosphere microbiome has only begun.

## Figures and Tables

**Table 1 microorganisms-10-01152-t001:** Examples of studies of bacterial and fungal rhizobiome diversity in *P. notoginseng*. The first most dominant phylum followed by the second and third most dominant phyla are listed with common genera within each phylum in parentheses when mentioned.

First	Second	Third	Ref.
Actinobacteria (*Actinoallomurus*, *Arthrobacter*)	Bacteroidetes (*Dyadobacter*, *Pedobacter*)	Proteobacteria (*Aquicella*, *Pseudomonas*, *Methylophilus*, *Pseudolabrys*)	[34]
Firmicutes (*Bacillus*, *Paenibacillus*, *Lysinibacillus*)	Actinobacteria (*Arthrobacter*, *Streptomyces*)	Proteobacteria (*Burkholderia*, *Cupriavidus*, *Pseudomonas*, *Dyella*)	[35]
Proteobacteria (*Pseudomonas*, *Rhodoplanes*)	Acidobacteria (*Candidatus solibacter*)	Actinobacteria (*Streptomyces*)	[36]
Ascomycota (*Fusarium*, *Dendryphion*, *Trichoderma*)	Basidiomycota (*Geminibasidium*, *Cryptococcus*)	Zygomycota (*Mortierella*)	[34]
Ascomycota (*Emericella*, *Fusarium*, *Plectosphaerella*, *Mycocentrospora*)	Zygomycota (*Mortierella*)	Basidiomycota (*Amanita*)	[37]
Zygomycota (*Mortierella*)	Ascomycota (*Fusarium*, *Phoma*)	Basidiomycota (*Cryptococcus*)	[38]

**Table 2 microorganisms-10-01152-t002:** Examples of studies of bacterial and fungal rhizobiome diversity in *P. ginseng*. The first most dominant phylum followed by the second and third most dominant phyla are listed with common genera within each phylum in parentheses when mentioned.

First	Second	Third	Ref.
Proteobacteria (*Sphingomonas*, *Paralcaligenes*)	Actinobacteria (*Frigoribacterium*)	Fusobacteria	[40]
Proteobacteria (*Pseudomonas*, *Sphingomonas*, *Pseudolabrys*)	Acidobacteria	Actinobacteria	[41]
Proteobacteria (*Pseudomonas*, *Burkholderia*)	Firmicutes (*Bacillus*)	-	[42]
Actinobacteria	Chloroflexi	Firmicutes	[43]
Proteobacteria	Actinobacteria	Acidobacteria/Chloroflexi	[44]
Ascomycota (*Fusarium*, *Tetrachaetum*)	Basidiomycota (*Cryptococcus*)	Zygomycota (*Mortierella*)	[40]
Ascomycota (*Fusarium*, *Alternaria*)	Basidiomycota	-	[43]
Ascomycota	Zygomycota	Basidiomycota	[44]

**Table 3 microorganisms-10-01152-t003:** Examples of studies of bacterial and fungal rhizobiome diversity in *P. quinquefolius*. The first most dominant phylum followed by the second and third most dominant phyla are listed with common genera within each phylum in parentheses when mentioned.

First	Second	Third	Ref.
Proteobacteria (*Kaistobacter*, *Rhodoplanes*, *Phenylobacterium*)	Actinobacteria (*Arthrobacter*, *Blastococcus*, *Mycobacterium*)	Acidobacteria (*Candidatus solibacter)*	[45]
Proteobacteria (*Bradyrhizobium*, *Pseudolabrys*)	Acidobacteria (*Candidatus solibacter*, *Bryobacter*)	Actinobacteria (*Streptomyces*, *Pseudoarthrobacter*)	[46]
Acidobacteria	Proteobacteria	Actinobacteria	[47]
Proteobacteria (*Rhodoplanes*, *Sphingomonas*, *Hyphomicrobium)*	Actinobacteria (*Pseudoclavibacter*, *Arthrobacter*, *Demequina)*	Acidobacteria	[48]
Ascomycota (*Penicillium*, *Didymella*, *Humicola*)	Zygomycota (*Mortierella*)	Basidiomycota (*Cystofilobasidium*)	[46]
Ascomycota	Basidiomycota	Chytridiomycota	[47]
Ascomycota (*Trichoderma*, *Penicillium*, *Acremonium*)	Rozellomycota (*Rozella*)	Zygomycota (*Mortierella*)	[48]

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
