# Peer review of "The Rhizosphere Microbiome of Ginseng"

_microorganisms, 2022, doi:10.3390/microorganisms10061152_

Round 1

Reviewer 1 Report

The article is well written and structured and dealt with a very valid topic.

L10, L82: Panax is a genus and not a species.

L24: I suggest to change this part of the sentence “microorganisms in plants”. In my opinion, the meaning could be misunderstood.

L55: Change “with” with for and “operational taxonomic units” with “Operational Taxonomic Units”.

L99: Add a comma after pH.

L106: Change “carbon/nitrogen (C/N) ratio” with “carbon/nitrogen ratio (C/N)”.

L244-L245: I did not understand what mean arched and flat shades.

L306-L307: Change “levels of Proteobacteria belonging to Paralcligenes and Sphingomonas” with “levels of Paralcligenes and Sphingomonas belonging to Proteobacteria”.

L473: Delete the “A” before B. amyloliquefaciens.

L473-L474: Add a comma after P. ginseng.

Author Response

L10, L82: Panax is a genus and not a species.

Have reworded to species of Panax.

L24: I suggest to change this part of the sentence “microorganisms in plants”. In my opinion, the meaning could be misunderstood.
Have changed to associated with plants.

L55: Change “with” with for and “operational taxonomic units” with “Operational Taxonomic Units”.

Changed.

L99: Add a comma after pH.

Changed.

L106: Change “carbon/nitrogen (C/N) ratio” with “carbon/nitrogen ratio (C/N)”.

Changed.

L244-L245: I did not understand what mean arched and flat shades.

I have added that arched and flat refers to the shape of the shades.

L306-L307: Change “levels of Proteobacteria belonging to Paralcligenes and Sphingomonas” with “levels of Paralcligenes and Sphingomonas belonging to Proteobacteria”.

Changed.

L473: Delete the “A” before B. amyloliquefaciens.
Have deleted "a" and added the strain designation.

L473-L474: Add a comma after P. ginseng.

Added comma and changed "that" to "which".

Reviewer 2 Report

The article contains detailed information about the cultivation of ginseng. The influence of different related factors on ginseng rhizosphere is generally described.

Important points regarding the phytopathogenic fungi abundance in the rhizosphere were discussed. However, the explanation of the phytopathogenic fungi abundance increase during cultivation is lacking.

Also, there are several comments that need minor changes:

Line 11 - should it be "plant" after ginseng? please check

Line 14 - I would not recommend to use "degradation" in this context, here should be something more like "disruption" or "change"

Line 14-16 - why are these microbes could improve the production? please add the information

Lines  34-61 - this part contain the information regarding different techniques of microbiome studying. I would recommend to shorten this part, especially because there was no or little discussion of the information obtained with classical methods.

Line 66 - reference regarding soil effect on rhizospere. there are only 3 references, I suggest that it should be more or at least a review about soil effects on rhizosphere without mentioning any exact plant

Line 84-85 - references to these studies are necessary

Line 101 - forest leaf litter adds more organic matter to the upper soil horizon, but not directly to the rhizospere; so it's necessary to correct this sentence

Line 113-114 - microbiome may be supressed by pesticide addition, but not reduced (while microbial biomass or activity may be reduced). It would be great to see some reference here

Line 117-118 - tillage typically reduce SOM - please add the reference, it's not obvious and true for all soils and types of tillage

Lines 123-125 - please clarify if the ginseng plants are fully removed during the harvesting or not. If they are removed, then it's no rhizosphere left.

Line 137 - " F11, Rb1, Rb2, Rc, Rd, Re and Rg1" what does these names mean? It's necessary to describe the differences between these ginsenoids if you mention it

Lines 149 - 217 - a detailed description of rhizobiomes of different Panax species. If possible, please include some information obtained with classical methods

Line 232-233 I would recommend to cite some review regarding the influence of fertilizers on rhizosphere microbiome

Line 288 - fied_Nectriaceae - should be italic?

Lines 328-344 - may be this part would be more suitable in the next chapter (5)? Or the chapter 5 should be renamed to " Impacts of ginseng root exudates on rhizosphere microbiome"

Author Response

The article contains detailed information about the cultivation of ginseng. The influence of different related factors on ginseng rhizosphere is generally described.

Important points regarding the phytopathogenic fungi abundance in the rhizosphere were discussed. However, the explanation of the phytopathogenic fungi abundance increase during cultivation is lacking.

While many articles note the increases in phytopathogenic fungi abundance increase during cultivation, there has not yet been a paper examining the mechanisms.

Also, there are several comments that need minor changes:

Line 11 - should it be "plant" after ginseng? please check

I am using "ginseng" as the plant name throughout the article rather than "ginseng plant".

Line 14 - I would not recommend to use "degradation" in this context, here should be something more like "disruption" or "change"

Have written as "negative changes".

Line 14-16 - why are these microbes could improve the production? please add the information

Sentence has been reworded to make it clearer that those beneficial microbes can improve production.

Lines  34-61 - this part contain the information regarding different techniques of microbiome studying. I would recommend to shorten this part, especially because there was no or little discussion of the information obtained with classical methods.

I would like to keep this section as is because I later describe different studies which have used different methods, and thus the reader will have some background about the limitations of the methods involved.

Line 66 - reference regarding soil effect on rhizospere. there are only 3 references, I suggest that it should be more or at least a review about soil effects on rhizosphere without mentioning any exact plant

I only chose 3 examples to show the effect of soil type, soil organic matter and plant genotype. This paragraph could be expanded into an entire review article, whereas it is just meant to introduce the concept of soil effect on the rhizospere in the introduction. The purpose is similar to introducing the issue about different microbiome techniques in the preceding paragraph.

Line 84-85 - references to these studies are necessary

I have reworded this sentence to just mention that there are many more studies recently, which is what the review is addressing. If I cited all the recent references for high-throughput sequencing in the introduction, then it would be a long list and repeating the citations where a description is provided for each study.

Line 101 - forest leaf litter adds more organic matter to the upper soil horizon, but not directly to the rhizospere; so it's necessary to correct this sentence

I have added that the leaf litter arrives to the upper soil horizon.

Line 113-114 - microbiome may be supressed by pesticide addition, but not reduced (while microbial biomass or activity may be reduced). It would be great to see some reference here

I have added a citation about reduced soil microbial populations with fumigation.

Line 117-118 - tillage typically reduce SOM - please add the reference, it's not obvious and true for all soils and types of tillage

I have added a citation about tillage typically reducing soil organic matter.

Lines 123-125 - please clarify if the ginseng plants are fully removed during the harvesting or not. If they are removed, then it's no rhizosphere left.

I have changed to soil microbiome from rhizosphere microbiome as there are some roots left but most are removed.

Line 137 - " F11, Rb1, Rb2, Rc, Rd, Re and Rg1" what does these names mean? It's necessary to describe the differences between these ginsenoids if you mention it

I have rewritten the sentence to say that these are specific ginsenoside compounds, and have made the same change in the following sentence.

Lines 149 - 217 - a detailed description of rhizobiomes of different Panax species. If possible, please include some information obtained with classical methods

It would take too much space to give detailed descriptions of the microbiome from each study, and the molecular studies provide the best overview of the microbes present. Some studies give large numbers of different taxa found, particularly in supplemental materials, whereas others provide a broader overview with a limited number of taxa described. Therefore, I have given the most common ones in table 1 from each study, so that they are comparable. This review is not designed to list all of the microbiomes found in ginseng soil, but to show common ones and to show whether they are similar between studies as well as between Panax spp. using similar methods.

Line 232-233 I would recommend to cite some review regarding the influence of fertilizers on rhizosphere microbiome

I have left ref#52 for this as an example. I could not find a review article about the influence of fertilizers on rhizosphere microbiome, but found a number of research articles, such as ref#52 that I cited.

Line 288 - fied_Nectriaceae - should be italic?

Have made it italics.

Lines 328-344 - may be this part would be more suitable in the next chapter (5)? Or the chapter 5 should be renamed to " Impacts of ginseng root exudates on rhizosphere microbiome"

I agree and have moved that paragraph to section 5 of the review.